# Generation and Purification of RANKL-Derived Small-Fragment Variants for Osteoclast Inhibition

**DOI:** 10.3390/pharmaceutics17111385

**Published:** 2025-10-25

**Authors:** Hyungjun Lee, Hyungseok Park, Kabsun Kim, Youngjong Ko, Chang-Moon Lee, Wonbong Lim

**Affiliations:** 1Department of Premedical Science, College of Medicine, Chosun University, Gwangju 61452, Republic of Korea; horsej423@naver.com (H.L.); kabsun@hanmail.net (K.K.); yeasts@hanmail.net (Y.K.); 2Laboratory of Orthopedic Research, Chosun University Hospital, Gwangju 61452, Republic of Korea; hyungseok929@chosun.ac.kr; 3Department of Orthopaedic Surgery, College of Medicine, Chosun University, Gwangju 61452, Republic of Korea; 4Regional Leading Research Center, Chonnam National University, Yeosu 59626, Republic of Korea; cmlee@chonnam.ac.kr; 5School of Healthcare and Biomedical Engineering, Chonnam National University, Yeosu 59626, Republic of Korea

**Keywords:** osteoporosis, osteoclast, receptor activator of NF-kappa B ligand (RANKL), HisTrap affinity chromatography

## Abstract

**Background/Objectives:** Osteoporosis is caused by excessive osteoclast activation via the receptor activator nuclear factor kappa B ligand (RANKL), which is released from osteoblasts or osteocytes. RANKL regulates osteoclast activity by binding to the receptor activator of nuclear factor kappa B (RANK) in the canonical pathway or leucine-rich repeat-containing G protein-coupled receptor 4 (LGR4) in the non-canonical pathway. In this study, we attempted to develop an intact small-fragment protein based on RANKL by removing the RANK-binding site and transforming the amino acid residues at crucial sites to inhibit osteoclast activity and treat osteoporosis. **Methods**: We expressed a small-fragment variant of RANKL as a soluble glutathione S-transferase (GST) or 6x histidine (His)-tagged fusion protein using a GST- or His-binding domain tag expression vector system. To generate an intact form of small-fragment RANKL, ribosome-inactivating protein–His-fusion RANKL was purified using HisTrap affinity chromatography and treated with tobacco etch virus nuclear inclusion endopeptidase to remove the His-tag fusion protein. Tartrate-resistant acid phosphatase (TRAP) and bone resorption pit formation assays were performed to analyze the inhibitory effects on osteoclast differentiation and activation. **Results:** The intact forms of 225RANKL295P and 225RANKL295A showed the strongest inhibitory effects on TRAP activity and bone resorption pit formation. **Conclusions:** Using an optimal construct design, a large and diverse range of small RANKL fragments could be generated. This suggests that the generation of small-fragment RANKL provides a promising avenue for the advancement of novel therapeutic approaches to osteoporosis.

## 1. Introduction

Receptor activator of nuclear factor kappa-B ligand (RANKL) is a member of the tumor necrosis factor (TNF) superfamily with a pivotal role in bone resorption by initiating and promoting osteoclast activation [1]. RANKL has several indicators, including the osteoprotegrin ligand, osteoclast differentiation factor, and TNF-related activation-induced cytokine [2]. It is a homotrimeric transmembrane protein mainly secreted by osteoblasts and osteocytes [3]. RANKL is initially expressed in a membrane-bound form and is cleaved by proteolytic enzymes, resulting in the release of its extracellular domain in a soluble form [4,5]. This soluble RANKL binds to the RANK (receptor activator of nuclear factor kappa-B) receptor on osteoclast precursor cells such as monocytes or macrophages and induces the initiation of the canonical nuclear factor kappa-B (NF-kB) pathway for osteoclast differentiation and activation [6]. In contrast, soluble RANKL binds to LGR4 (leucine-rich repeat-containing G-protein-coupled receptor 4) and triggers the negative regulation of osteoclast activation [7,8]. Notably, LGR4 signaling from RANKL–LGR4 binding is known to promote glycogen synthase kinase-3 beta (GSK-3beta) phosphorylation and inhibit the translocation of NFATc1 for osteoclast-related gene transcription, thereby helping to alleviate osteoclast overactivation [9]. Therefore, strategies to accelerate RANKL–LGR signaling in osteoclast precursor cells may effectively modulate excessive osteoclast differentiation and activation during severe bone resorption, influencing the blockade of RANKL–RANK signaling [10,11].

In a previous report, experiments were performed using LGR4-ECD or mutant RANKL with a transformed amino acid residue at the RANK–RANKL binding site to determine whether targeting LGR4 could decrease RANKL-induced osteoclast activation [7,11]. This pharmaceutical approach may be an effective strategy for osteoporosis treatment based on the compensatory regulation of osteoclast activity via LGR4, regardless of soluble RANKL overactivation [12]. In another study, a RANKL-derived small protein with the RANK binding site removed as a ligand for LGR4 inhibited osteoclast activation [13,14]. The cleaved protein 225RANKL295 from full-length RANKL was shown to inhibit osteoclast differentiation activity and reduce the binding affinity of 225RANKL295 to RANK while maintaining its affinity for LGR4 [15]. Furthermore, amino acid sequence 295 is a critical point for LGR4 binding, and its transformation to a different amino acid should be investigated to improve the binding affinity of LGR4.

The small fragment size of proteins has several advantages. Compared to antibodies and other biologics, peptide-sized proteins may offer a more favorable option for treating chronic diseases due to their distinct advantages, including cost-effectiveness and the ability to induce the desired effect with small-protein doses [16,17]. Most importantly, a smaller peptide size is preferred to reduce immunogenicity in therapeutic applications, as longer protein sequence chains face a higher possibility of elimination by neutralizing antibodies [18,19]. Therefore, the present study provides a protocol for screening small fragments of RANKL-based proteins during osteoclast inhibition. The small-fragment proteins were expressed in *E. coli*, and high-purity proteins were produced through a simple process using a chromatography column [20,21]. To improve the expression and purification of these modified small fragments of RANKL, tagging proteins such as soluble glutathione S-transferase (GST) and 6x histidine (His) were removed by the tobacco etch virus (TEV) site [22]. Fusion tags are indispensable tools for protein expression and purification in bacteria [23]. However, the presence of a fusion tag may interfere with protein function, and its removal from the target protein is desirable [24]. TEV protease is one of the most widely used enzymes that removes fusion tags from recombinant proteins due to its stringent sequence specificity [25]. However, TEV protease may require a Gly or Ser residue at the C-terminus (P1′ position) of its recognition site, leaving a non-native Ser or Gly residue at the N-terminus of the target protein after tag removal [26].

Therefore, in this study, we aimed to produce a small fragment of RANKL in an intact form without a tagging protein and to improve the protein by point mutation of the 295th amino acid, which is expected to play a key role in binding to LGR4. The results present a process for purifying the RANKL fragment in an intact form and generating an optimized small-fragment variant of RANKL, suggesting the potential of this RANKL variant as a new therapeutic agent for osteoporosis. Further research will allow for the design of an optimized protein.

## 2. Materials and Methods

### 2.1. Construction of a GST-Tagged RANKL Variant Expression Vector

All reagents were purchased from Sigma-Aldrich (St. Louis, MO, USA). The DNA template encoding the amino acid sequence of mouse RANKL was synthesized using BIONEER (Daejeon, Republic of Korea). The amplification and cloning of the RANKL fragment were performed as described in our previous study. Briefly, the polymerase chain reaction (PCR) product was cloned into the BamHI/XhoI sites of the pGEX-4T-1 vector (Promega, Madison, WI, USA) and transformed into Escherichia coli BL21-gold competent cells (Agilent, Santa Clara, CA, USA) via electroporation (5 ms, 12.5 kV/cm). The transformed *E. coli* cells were cultivated in Luria–Bertani (LB) broth with ampicillin (50 μg/mL, T&I, Daejeon, Republic of Korea). The cloned product was confirmed by a commercial sequencing service (SolGent Co., Daejeon, Republic of Korea). All sequence data were analyzed using Vector NTI Advance9.1.0 (Invitrogen, Carlsbad, CA, USA). The transformed *E. coli* were cultured at 37 °C with shaking at 200 RPM until the optical density (OD) at 600 nm reached 0.8–1.0. Then, protein expression was induced by adding isopropyl β-D-1-thiogalactopyranoside (IPTG) to a final concentration of 0.5 mM. After induction, the bacteria were cultured at 18 °C and 200 RPM for 24 h.

### 2.2. Initial Chromatography for the Purification of GST-Tagged RANKL Variants

Cultured bacteria were harvested by centrifugation for 10 min at 4 °C and 6000× *g*. After centrifugation, the bacterial pellet was collected and resuspended in 30 mL of phosphate-buffered saline (PBS) per 3 g of wet bacterial pellet. The resuspended bacteria were treated with sarcosyl (0.3%), β-mercaptoethanol (6.673 mM), lysozyme (100 μg/mL), and phenylmethylsulfonyl fluoride (1 mM). Samples were disrupted using an ultrasonicator and centrifuged at 25,029× *g* for 30 min. Following centrifugation, the supernatant was collected and filtered using a 0.45 μm filter. Afterward, the sample was loaded onto a 5 mL GST Trap HP column equilibrated with 20 mL of PBS using fast-protein liquid chromatography (FPLC, Cytiva, Marlborough, MA, USA). The column was washed with PBS (10 mL) and eluted with a glutathione buffer (5 mM). The eluted GST-tagged RANKL variants were loaded onto a 10% sodium dodecyl sulfate (SDS)–polyacrylamide gel. After separation, the gel was stained with Coomassie brilliant blue G-250, and images were acquired using a digital scanner (EPSON, Los Alamitos, CA, USA).

### 2.3. Three-Dimensional Structure Simulation of a Small RANKL-Derived Protein

The full-length RANKL sequence was searched against amino acid sequences in the Protein Data Bank using BLAST (http://blast.ncbi.nlm.nih.gov, accessed on 17 February 2025). The simulation of the 3D structures of RANKL-derived fragments and a template search were performed using Phyre2 (http://www.sbg.bio.ic.ac.uk/~phyre2/html/page.cgi?id=index, accessed on 17 February 2025).

### 2.4. Biological Activity Test of Purified RANKL Variants

All cell-based studies involving live mice were performed in accordance with institutional and governmental requirements and were approved by the Institutional Animal Care and Use Committee (approval no. CIACUC2021—S0007). For the tartrate-resistant acid phosphatase (TRAP) assay, bone marrow macrophages (BMMs) were flushed from the femurs of 6-week-old female Balb/c mice and cultured in alpha-MEM containing 10% fetal bovine serum (Thermo Fisher Scientific Inc., Waltham, MA, USA) and 30 ng/mL macrophage colony-stimulating factor (M-CSF; R&D Systems, Minneapolis, MN, USA) after treatment with ACK lysis buffer (Gibco, Gaithersburg, MD, USA). The BMMs were plated at a density of 1.4 × 10^4^ cells/well in 96-well plates, and cells were cultured at 37 °C in 5% CO_2_ for 4 d. Subsequently, the cells were fixed with 10% formalin for 20 min and stained with a TRAP solution for 30 min. Multinucleated osteoclasts were observed using an ECLIPSE Ts2R microscope (Nikon, Tokyo, Japan). The ratio of the resorbed to total area was calculated using ImageJ analytical software (Version 1.54k http://rsbweb.nih.gov/ij/ (accessed on 17 February 2025); National Institutes of Health, Bethesda, MD, USA).

### 2.5. Construction of His-Tagged RANKL Variant Expression Vector

The 157G–316D amino acid sequence of mouse RANKL was cloned into the NdeI/XhoI site of the pET-30a (+) expression vector (Promega) and transformed into *E. coli* BL21-gold competent cells (Agilent) via electroporation (5 ms, 12.5 kV/cm).

To construct a RANKL-based small-protein modification, the template DNA encoding the G225RANKLP295 amino acid sequence of mouse RANKL was amplified using the 157RANKL316 DNA template and the primers listed in Table 1. This vector was applied to *E. coli*-BL21 (DE3), and cells were cultured in 1000 mL of LB medium in a 2000 mL baffled flask at 37 °C and 200 RPM. IPTG was added to achieve a final concentration of 0.2 mM. After induction, the bacteria were cultured at 16 °C and 200× *g* for 24 h.

### 2.6. Initial Chromatography for the Purification of the RANKL Variant Using His Trap Affinity Chromatography

Cultured *E. coli* were harvested by centrifugation for 10 min at 4 °C and 6000× *g*. The pellets were collected and resuspended in 50 mL of 1 mM imidazole buffer containing 25 mM Tris, 500 mM NaCl, and 1 mM imidazole, and adjusted to pH 8.0 using distilled water (DW). The resuspended bacteria were treated with 1 mM phenylmethylsulfonyl fluoride, disrupted by sonication, and centrifuged at 25,029× *g* for 30 min. After centrifugation, the supernatant was collected to isolate soluble proteins and filtered using a 0.45 μm filter. Then, the sample was loaded onto a 5 mL HisTrap HP column equilibrated with 20 mL of 1 mM imidazole buffer using FPLC (ÄKTA FPLC explorer system, Amersham Pharmacia Biotech, Stockholm, Sweden). The column was washed with 10 mL of the 1 mM imidazole buffer and subsequently eluted with 20 mL of 100 and 500 mM imidazole buffers (25 mM Tris, 500 mM NaCl, 100 mM/500 mM imidazole, pH 8.0). The eluted His-tagged RANKL variants were loaded onto a 10% SDS–polyacrylamide gel, and fractions containing His-tagged RANKL variants were collected using SDS–polyacrylamide gel electrophoresis (SDS-PAGE).

### 2.7. Secondary Chromatography for the Purification of RANKL Variants Using Size-Exclusion Chromatography

The purified His-tagged RANKL variants were injected into a 5 mL tube and loaded onto a Hi-Load 16/600 Superdex 200 pg column (Cytiva) equilibrated with 252 mL of a size-exclusion chromatography (SEC) buffer (50 mM sodium phosphate, 150 mM NaCl, and pH 7.4 in DW). Each sample was subsequently eluted with a 180 mL SEC buffer, and the first eluted peak was collected. The eluted His-tagged RANKL variants were analyzed using SDS-PAGE.

### 2.8. Cleavage of His-Tag by TEV Enzyme and Tertiary Chromatography for the Purification of the Unbound Fraction in RANKL Variants

His-tagged RANKL variants were cleaved using TEV protease. The cleaved product was prepared by incubating 1 μg of TEV protease per 50 μg of target protein in SEC buffer (50 mM sodium phosphate, 150 mM NaCl, pH 7.4) at 30 °C for 24 h without shaking. Subsequently, the unbound fraction of the RANKL variants was purified using His-tag affinity chromatography. N-Lauroylsarcosine sodium salt was added at a final concentration of 0.2% to prevent the aggregation of TEV-cleaved RANKL variants. The sarkosyl-treated sample was loaded onto a 5 mL His Trap HP column and equilibrated with 20 mL of 1 mM imidazole buffer (containing 0.2% sarkosyl) using FPLC. The column was washed with 10 mL of a 0 mM imidazole buffer containing 0.2% sarkosyl and subsequently eluted with 20 mL of an imidazole buffer at concentrations of 50, 300, and 500 mM, each containing 0.2% sarkosyl. The eluted fractions of His-tagged RANKL were analyzed via SDS-PAGE.

### 2.9. SDS-PAGE

A 15% running gel was prepared using a mixture of 30% acrylamide, 1.5 M Tris (pH 8.8), 10% SDS, 0.1% ammonium persulfate (APS), and 0.01% tetramethylethylenediamine (TEMED). The mixtures were poured into a prepared glass mold and allowed to solidify. The protein samples were prepared at a total volume of 40 μL using 5× SDS and reacted at 100 °C for 5 min prior to use. Electrophoresis was conducted at 100 V until the desired position was reached. Gels were stained with the EZ-Gel Staining Solution and photographed using a Gel Doc Go Imaging System (Bio-Rad, Hercules, CA, USA).

### 2.10. Bone Resorption Pit Formation Assay

To assess bone resorption pit formation, BMMs were seeded onto osteoassay plates coated with thin calcium phosphate films (Corning Inc., Tewksbury, MA, USA). BMMs were then incubated with M-CSF and RANKL for 5 d until mature osteoclasts resorbed the calcium phosphate film. After the culture period, the cells were dissolved in 5% sodium hypochlorite, and images of the resorption pits were acquired using a light microscope (Nikon). The ratio of the resorbed area to the total area was calculated using ImageJ software.

### 2.11. Statistical Analysis

All experiments were performed in triplicate. Data were expressed as the mean ± SD and analyzed via analysis of variance (ANOVA) using SPSS for Windows (version 12.0; SPSS, Chicago, IL, USA) to determine significant differences.

## 3. Results

### 3.1. GST-Tagged 158RANKL316 and 225RANKL295 Proteins

A schematic representation of the GST-tagged RANKL variants (158RANKL316-GST and 225RANKL295-GST) is shown in Figure 1A. A modified pMX vector containing 158RANKL316 or 225RANKL295 was used to produce the expression vector pMX-RANKL variant for soluble GST-fusion RANKL. To confirm the nucleotide sequences of the cloned GST-fused RANKL, DNA sequencing analysis was performed, and the obtained sequences were compared with theoretical nucleotide sequences for the TNF domain of murine RANKL. The nucleotide sequences of the RANKL variants were identical to those of the murine RANKL TNF domain. Using FPLC, 158RANKL316-GST and 225RANKL295-GST were purified from the transformed *E. coli*.

To determine the minimum size of the RANKL-derived protein required for osteoclast inhibition, small RANKL fragments ranging from 225 to 295 amino acid residues were synthesized. The expression of each small RANKL fragment cloned in the pMX vector containing 158RANKL316 or 225RANKL295 was induced with IPTG in *E. coli* BL21-Codon Plus (DE3)-RIPL. The recombinant proteins 158RANKL316-GST and 225RANKL295-GST were purified using column chromatography (Figure 1B). SDS-PAGE revealed bands of approximately 47 kDa (158RANKL316-GST) and 36 kDa (225RANKL295-GST) in the total cellular protein extracts of the induced samples, matching the predicted size of the product expressed from our construct (Figure 1C). GST-tagged 158RANKL316 and 225RANKL295 were analyzed via a 3D simulation (Figure 1D). The GST region did not overlap with the 158RANKL316 or 225RANKL295 regions in the 3D simulation. Both 158RANKL316-GST and 225RANKL295-GST did not affect the BMMs viability, respectively (Appendix A). BMMs differentiated into mature TRAP-positive multinucleated osteoclasts, and the number of mature osteoclasts was significantly higher in cells treated with 158RANKL316-GST alone. However, treatment with 225RANKL295-GST alone did not induce osteoclast differentiation into TRAP-positive osteoclasts (Figure 1E). In addition, 225RANKL295-GST had an inhibitory effect on the generation of TRAP-positive multinucleated cells in the presence of 158RANKL316 (Figure 1F).

### 3.2. Generation of 157RAKL316 and Effects on Osteoclasts

In accordance with a previous study, the 158RANKL316 sequence was chosen as the TNF superfamily homology site for osteoclast activity, and the TEV cleavage sequence was induced between the 6xHis-tag and RANKL to purify the intact form of the protein.

A schematic diagram of 6xHis-157RANKL316, utilized as a control in the TRAP assay for RANKL-derived small proteins modified by point mutations, is shown in Figure 2A. A DNA template encoding the amino acid sequence from Gly157, which is easily cleaved by TEV, to Asp316 in mouse RANKL was synthesized. A NdeI_6His_TEV cleavage sequence and restriction enzyme sites for XhoI were inserted at the 5′ and 3′ ends, respectively. To investigate the effect of 157RANKL316 on osteoclast differentiation, a TRAP assay was performed using the purified 157RANKL316 protein (Figure 2B). A significant dose-dependent increase in osteoclast differentiation was observed when 157RANKL316 was administered alone (Figure 2C).

### 3.3. Generation and Purification of 225RANKL295P and Modified Proteins

A schematic diagram of 225RANKL295P and modified small fragments of RANKL (225RANKL295G, 225RANKL295A, 225RANKL295V, and 225RANKL295L) at the 295th amino acid point modified to the same residue group is shown in Figure 3A. To enhance this inhibitory effect, point mutations were engineered to modify proline (P-Pro) at position 295, which is predicted to be a critical site for LGR4 binding. These mutations replaced proline (Pro) with the non-polar amino acids glycine (Gly), alanine (Ala), valine (Val), and leucine (Leu). To enable the purification of these modified small fragments of RANKL without the tag protein, a TEV cleavage site and 224S, required for TEV cleavage, were designed to be inserted upstream of 225E according to 225RANKL295P-His (Figure 3B). The intact form of 225RANKL295P and the modified small fragments of RANKL (225RANKL295G, 225RANKL295A, 225RANKL295V, and 225RANKL295L) were determined via a 3D simulation (Figure 3C). The overall structure of 225RANKL295P and the modified small fragments of RANKL were observed to be similar in 3D simulations. Initial purification was conducted using His-tag affinity FPLC, and further purification was performed using SEC via FPLC, yielding protein fractions of appropriate size (Figure 3D). Similar peaks were observed for each protein; the generated proteins were confirmed by SDS-PAGE (Figure 3E). His-tag cleavage was performed using TEV protease, followed by a second His-tag FPLC purification step to remove the His-tag, resulting in the isolation of 225RANKL295P and the modification of other small fragments of RANKL (Figure 3F). The final products were confirmed using SDS-PAGE (Figure 3G).

### 3.4. TRAP Analysis and Bone Resorption by Modified Proteins

Finally, to determine the protein with the strongest inhibitory effect on osteoclastogenesis, a TRAP assay was performed using the purified protein candidates, with 157RANKL316 serving as a control (Figure 4A,B). All five candidates, including 225RANKL295P, inhibited osteoclast differentiation. Among the candidates, 225RANKL295V showed the weakest inhibitory effect, whereas 225RANKL295P and 225RANKL295A had the strongest overall effect. Based on these results, bone resorption assays were performed to confirm the inhibitory effects of 225RANKL295P and 225RANKL295A on osteoclast activity (Figure 4C,D). Numerous resorption pits were observed in mature osteoclasts treated with 158RANKL316; however, treatment with 225RANKL295P or 225RANKL295A significantly decreased the area of the resorption pits. These results indicate that 225RANKL295P and 225RANKL295A are potential inhibitors of osteoclast activity.

## 4. Discussion

The development of inhibitors targeting the RANKL–RANK signaling pathway may be a viable therapeutic strategy for treating osteoporosis [27]. LGR4 was recently identified as an additional receptor for RANKL that competes with RANK for RANKL binding, thereby inhibiting RANK signaling and osteoclast activation [13]. To investigate this concept in the present study, a small RANKL-derived fragment was used as a GST-fusion protein to inhibit RANKL–RANK signaling according to a previous study [15]. This small-fragment protein was synthesized using the RANKL sequence and exhibited a structure that closely resembled the original RANKL. A specific small fragment, 225RANKL295-GST, significantly inhibited osteoclast differentiation and activation. Treatment with the small RANKL fragment effectively hindered osteoclast differentiation in the presence of active RANKL, suggesting that 225RANKL295-GST functions as a competitive inhibitor of RANKL. However, the GST-tag is relatively large (26 kDa) and forms a dimer on its own, which increases the probability of scavenging antigen–antibody reactions [28]. Therefore, it was necessary to remove the GST tag from the original protein and purify it.

A previous study stated that treatment with a recombinant protein based on the active part of RANKL only led to prompt osteoclast differentiation [29]. This suggests that the intact form of RANKL constructed at the TNF superfamily homology site can induce osteoclast differentiation and may serve as an alternative to full-length RANKL.

A His-tag was incorporated to facilitate purification, and the ribosome inactivation protein (RIP) was employed to enhance protein expression [30]. RIP has several biological properties, and its expression increases under stressful conditions [31]. Several possible applications of RIP in medicine involve linking or fusing it with antibodies or other appropriate carriers to form immunotoxins or other conjugates that are specifically toxic to harmful cells [32].

Functional protein production in mammalian cells has been known to have disadvantages including high costs and viral contamination [33]. In contrast, the *E. coli* expression system is low in cost with a high yield, though it is usually expressed as an inclusion body [34]. This problem can be solved using diverse fusion-tagged proteins. In the present study, 6xHis-RIP-TEV-RANKL variants were generated from a modified pET-30a (+) expression vector containing the 6xHis-RIP fusion and a TEV protease cleavage site. Thus, we expressed RANKL variants in a soluble fusion form and purified the RANKL fragment expressed using a His-tag affinity column. To isolate the 6xHis-fusion RANKL variant, a His-tag affinity column was used, and the intact proteins were acquired to remove the 6xHis-RIP from the RANKL variants prior to the second instance of His Trap affinity column purification after TEV protease cleavage. Secondary His Trap affinity chromatography was performed to purify the cleaved RANKL. Two-step His-Trap affinity chromatography was very effective and simple for the purification of RANKL variants.

It is expected that the optimized design of small-fragment RANKL variants will become feasible through continued research. Our results provide empirical support for the potential use of RANKL-derived variants as therapeutic interventions for osteoporosis.

## 5. Conclusions

The current approach, which employs the targeted integration of an immunogenic amino acid into a protein, may serve as a viable solution to circumvent these challenges. In this study, we showed that 225RANKL295P and 225RANKL295A, small fragments derived from RANKL based on its protein structure, inhibit TRAP and osteoclast activity. The findings of this study suggest that the overall purification process of RANKL-derived small proteins is a promising avenue for advancing novel therapeutic approaches in the treatment of osteoporosis.

## Figures and Tables

**Figure 1 pharmaceutics-17-01385-f001:**
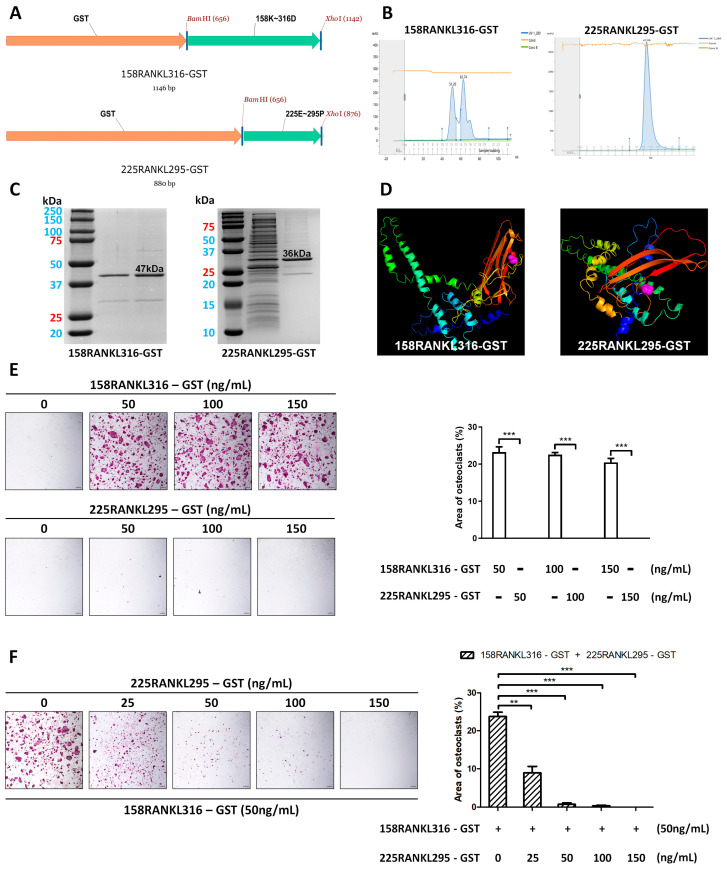
Generation of RANKL-derived small proteins 158RANKL316-GST and 225RANKL295-GST. (**A**) Schematic diagram of 158RANKL316-GST (**above**) and 225RANKL295-GST (**below**). (**B**) The purified proteins of 158RANKL316-GST (**left**) and 225RANKL295-GST (**right**) were obtained from the 500 mM imidazole elution peak after FPLC purification. (**C**) SDS-PAGE results of 158RANKL316-GST (**left**) at 47 kDa and 225RANKL295-GST (**right**) at 36 kDa. (**D**) Three-dimensional simulation of 158RANKL316-GST (**left**) at 47 kDa and 225RANKL295-GST (**right**). (**E**) **Left**: Dose-dependent effect of 158RANKL316-GST or 225RANKL295-GST on the formation of TRAP-positive multinucleated cells (40× magnification, 50 μm scale bar). **Right**: Quantification of TRAP-positive osteoclasts, as determined using ImageJ. The error bars represent the mean ± SD from three independent experiments. When comparing quantified data, significance is indicated at *** *p* < 0.001. (**F**) **Left**: The effect of the 225RANKL295-GST on the formation of TRAP-positive multinucleated cells in the presence of 158RANKL316-GST (50 ng/mL) in a dose-dependent manner. (40× magnification, 50 μm scale bar). **Right**: Quantification of TRAP-positive osteoclasts, as determined using ImageJ. The error bars represent the mean ± SD from three independent experiments. When comparing quantified data, significance is indicated at ** *p* < 0.01, *** *p* < 0.001.

**Figure 2 pharmaceutics-17-01385-f002:**
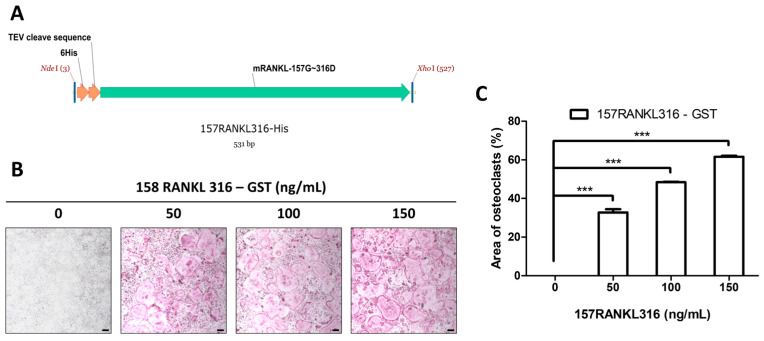
Generation of the RANKL-derived small protein 158RANKL316. (**A**) Schematic diagram of 157RANKL316. (**B**) Dose-dependent effect of 157RANKL316 on the formation of TRAP-positive multinucleated cells (40× magnification, 50 μm scale bar). (**C**) Quantification of TRAP-positive osteoclasts, determined using ImageJ. The error bars represent the mean ± SD from three independent experiments. When comparing quantified data, significance is indicated at *** *p* < 0.001.

**Figure 3 pharmaceutics-17-01385-f003:**
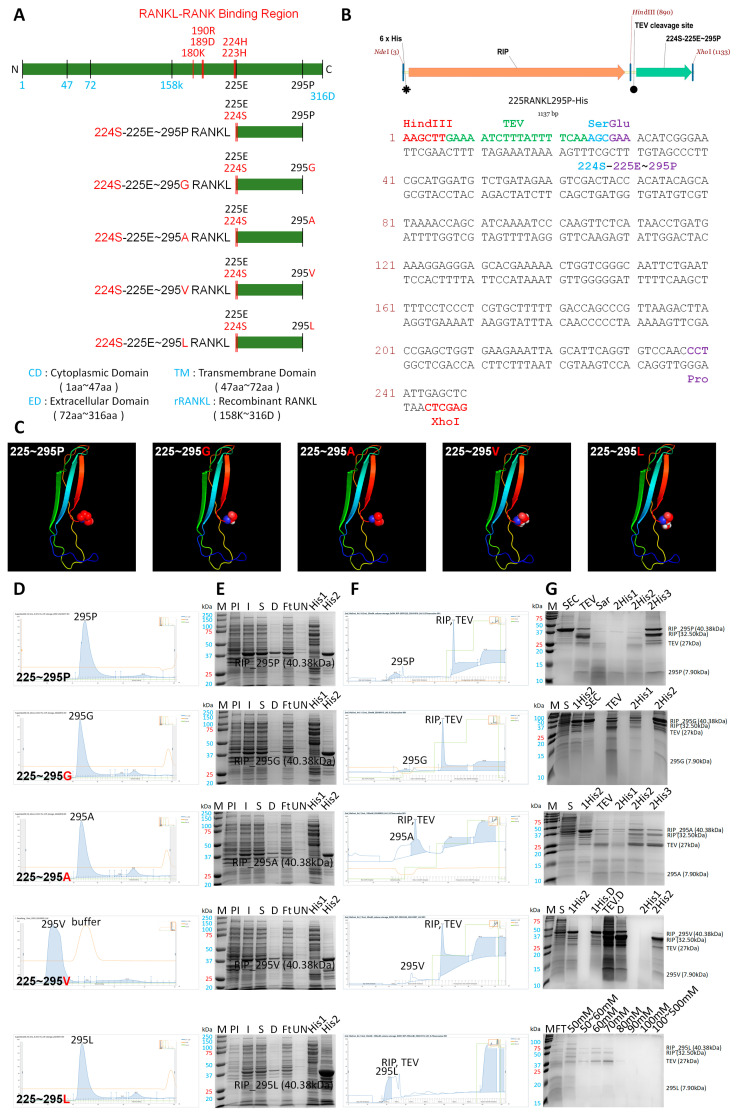
Generation of RANKL-derived small-protein 225RANKL295 RANKL variants. (**A**) Sequence of RANKL-derived small-protein 225RANKL295 RANKL variants with point mutations introduced using the non-polar amino acids glycine (G), alanine (A), valine (V), and leucine (L) at the 295P site of the 225RANKL295 construct. The 224th amino acid was transformed into serine for TEV cleavage. (**B**) Schematic diagram of the 6xHis-RIP-TEV cleavage site of 225RANKL295P. (**C**) Three-dimensional simulation of 225RANKL295P (225–295P), 225RANKL295G (225–295G), 225RANKL295A (225–295A), 225RANKL295V (225–295V), and 225RANKL295L (225–295L). (**D**) The protein was isolated from the initial peak using SEC. (**E**) The purification process was validated, and consistency with the intended protein was confirmed using SDS-PAGE. M, marker; PI, pre-induction; I, IPTG induction; S, supernatant; D, debris; Ft, flow through; UN, unbound washing; His1, HisTrap elution 1st peak; His2, HisTrap elution 2nd peak. (**F**) Purification by FPLC using the second His-tag following TEV cleavage. The purified protein was obtained from the elution peak with 50 mM imidazole. (**G**) The purified protein following TEV cleavage was confirmed through SDS-PAGE. SEC, size-exclusion chromatography; TEV, TEV cleavage; Sar, treatment with 0.2% sarcosine; 2His1, 2nd HisTrap elution 1st peak; 2His2, 2nd HisTrap elution 2nd peak; 2His3, 2nd HisTrap elution 3rd peak; 1HisD, 1st HisTrap-eluted debris; TEVD, TEV cleavage debris; 50 mM, 50 mM imidazole-eluted peak; 50~60 mM, 50–60 mM imidazole-eluted peak; 60 mM, 60 mM imidazole-eluted peak; 70 mM, 70 mM imidazole-eluted peak; 80 mM, 80 mM imidazole-eluted peak; 90 mM, 90 mM imidazole-eluted peak; 100 mM, 100 mM imidazole-eluted peak; 100~500 mM, 100–500 mM imidazole-eluted peak.

**Figure 4 pharmaceutics-17-01385-f004:**
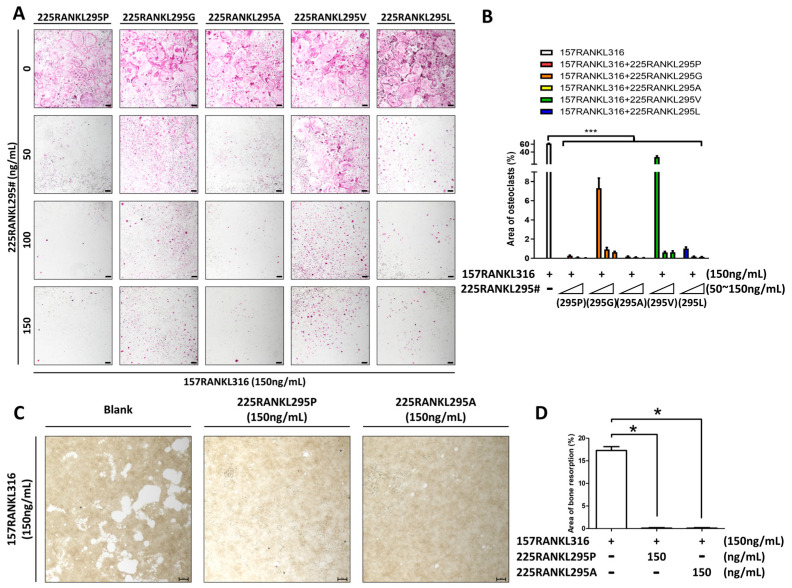
Effects of RANKL-derived small-protein 225RANKL295 RANKL variants on osteoclast differentiation and activation. (**A**) Dose-dependent effect of 225RANKL295P, 225RANKL295G, 225RANKL295A, 225RANKL295V, and 225RANKL295L in the presence of 157RANKL316 (150 ng/mL) (40× magnification, 50 μm scale bar). (**B**) Quantification of TRAP-positive osteoclasts, as determined using ImageJ. The error bars represent the mean ± SD from three independent experiments. When comparing quantified data, significance is indicated at *** *p* < 0.001. (**C**) Bone resorption assay of 225RANKL295P and 225RANKL295A in the presence of 157RANKL316 (150 ng/mL) (40× magnification, 50 μm scale bar). (**D**) Quantifications of resorption pit area, as analyzed using ImageJ. The error bars represent the mean ± SD from three independent experiments. When comparing quantified data, significance is indicated at * *p* < 0.05.

**Table 1 pharmaceutics-17-01385-t001:** Primers for PCR of the 295P point mutation.

Oligo	Sequence (5′ --> 3′)	Note
224S-5′	AAGCTTGAAAATCTTTATTTTCAAAGCGAAACATCGGGAAGCGTACCTACAG	S-225E
P295P-3′	CTCGAGTTACGGGTTGGACACCTGAATGCTAATTTCT	Non-polar
P295G-3′	CTCGAGTTAGCCGTTGGACACCTGAATGCTAATTTCT	Non-polar
P295A-3′	CTCGAGTTACGCGTTGGACACCTGAATGCTAATTTCT	Non-polar
P295V-3′	CTCGAGTTACACGTTGGACACCTGAATGCTAATTTCT	Non-polar
P295L-3′	CTCGAGTTACAGGTTGGACACCTGAATGCTAATTTCT	Non-polar

## Data Availability

Data presented in this study is contained within the article. Further inquiries can be directed to the corresponding author.

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
