# Peer review of "Generation and Purification of RANKL-Derived Small-Fragment Variants for Osteoclast Inhibition"

_pharmaceutics, 2025, doi:10.3390/pharmaceutics17111385_

Round 1

Reviewer 1 Report

Comments and Suggestions for Authors

This study explores a novel strategy for osteoporosis treatment by engineering small-fragment variants of RANKL that lack the RANK-binding site and contain amino acid substitutions to inhibit osteoclast activity. The researchers expressed these fragments as GST- or His-tagged fusion proteins, purified them, and removed the tags to obtain intact forms. Functional assays demonstrated that the 225RANKL295P and 225RANKL295A variants exhibited the strongest inhibition of osteoclast differentiation and bone resorption. These findings highlight the potential of small-fragment RANKL as a therapeutic platform for osteoporosis, although a couple of questions should be addressed to strengthen the study.

Major:

  1. Recommend testing recombinant proteins to determine whether they affect cell viability.

  1. The authors claim that the modified proteins inhibit osteoclast-mediated bone resorption. To substantiate this claim, fully differentiated osteoclasts should be used, followed by treatment with the modified proteins.

Minor:

  1. 1A and B

I recommend redrawing the schematic diagram for readers.

  1. In result 3.1

To generate GST-fusion proteins, what vector is used? pMX or pGEX?

  1. I recommend merging E and F, G and H. Add a statistic comparing 50, 100, and 150 ng/mL results.
  2. Typos, result 3.2 title, Page7 and recommend reviews the manuscripts to check typos
  3. Cell pictures, scale bar not noticeable.
  4. 3A and B, I recommend redrawing the schematic diagram for readers.

Author Response

Major:

1. Recommend testing recombinant proteins to determine whether they affect cell viability.

Thank you for kind comment. According to reviewer’s advice, we performed the cell viability assay in 158RANKL316-GST or 225RANKL295-GST-treated BMM by dose dependent manners and added the data as a supplementary materials.

2. The authors claim that the modified proteins inhibit osteoclast-mediated bone resorption. To substantiate this claim, fully differentiated osteoclasts should be used, followed by treatment with the modified proteins.

Thank you for kind comment. In Fig,4C, we showed the inhibitory effect of activation and differentiation of osteoclast by modified protein as reviewer commented. This analytical method is a typical method for simultaneously demonstrating osteoclast differentiation and activity, and has been validated in numerous references. It initiates the culture of BMMs on calcium plates, and can simultaneously demonstrate osteoclast differentiation and activity.

Minor:

1. 1A and B I recommend redrawing the schematic diagram for readers.

Thank you for kind comment. According to reviewer’s advice, we redrew the 1A and 1B to make clear.

2. In result 3.1 To generate GST-fusion proteins, what vector is used? pMX or pGEX?

In the result 3.1, we generated the GST-fusion protein using by pGEX-4T-1 vector. And It is commented at Material and Method section 2.1.

3. I recommend merging E and F, G and H. Add a statistic comparing 50, 100, and 150 ng/mL results.

Thank you for kind comment. According to reviewer’s advice, we merged the 1E-1F and 1G-1H panels. We also added a 225RANKL295-GST (0 ng/mL) condition to the 158RANKL316-GST + 225RANKL295-GST results (formerly 1G/1H) and included statistical comparisons between 0 ng/mL and other concentrations.

4. Typos, result 3.2 title, Page7 and recommend reviews the manuscripts to check typos

Thank you for kind comment. We checked it and corrected.

5. Cell pictures, scale bar not noticeable.

Thank you for kind comment. According to reviewer’s advice, we corrected the size bar.

6. 3A and B, I recommend redrawing the schematic diagram for readers.

Thank you for kind comment. According to reviewer’s advice, we redrew the 3A and 3B to make clear.

Reviewer 2 Report

Comments and Suggestions for Authors

This manuscript presents the design and development of engineered small-fragment RANKL proteins that lack the RANK-binding site but retain the ability to inhibit osteoclast activity, offering a new protein-based therapeutic strategy for osteoporosis. The study is innovative and well-executed in terms of protein expression and purification. However, two major concerns regarding experimental controls and mechanistic validation must be addressed to strengthen the findings.

Major Concerns:

  1. The study demonstrates that the 158RANKL316 fragment promotes osteoclast differentiation, whereas the 225RANKL295 fragment inhibits it. To ensure the inhibitory effects are not specific to the custom 158RANKL316 construct, the authors should repeat key experiments (TRAP and bone resorption assays) using standard recombinant RANKL (e.g., commercial soluble RANKL) to induce differentiation and test the inhibitory effects of 225RANKL295 variants under these conditions.
  1. The proposed mechanism of inhibition via LGR4 interaction is compelling but lacks direct evidence. The authors should assess the binding affinity of 225RANKL295 variants to LGR4 (e.g., using SPR or ELISA) to confirm specificity and rule out other potential pathways. This is critical to validate the mechanistic hypothesis.

Minor Concern:

  1. In 3D simulations (Fig. 1d, 3c), label key residues (e.g., 295 position) to enhance clarity and highlight the structural basis of the mutations.

Author Response

Major Concerns:

1. The study demonstrates that the 158RANKL316 fragment promotes osteoclast differentiation, whereas the 225RANKL295 fragment inhibits it. To ensure the inhibitory effects are not specific to the custom 158RANKL316 construct, the authors should repeat key experiments (TRAP and bone resorption assays) using standard recombinant RANKL (e.g., commercial soluble RANKL) to induce differentiation and test the inhibitory effects of 225RANKL295 variants under these conditions.

Thank you for your kind comment. The 158RANKL316 fragment were perfect matched with commercial soluble RANKL (R&D Systems) and we confirmed the activity in the previous study.

2. The proposed mechanism of inhibition via LGR4 interaction is compelling but lacks direct evidence. The authors should assess the binding affinity of 225RANKL295 variants to LGR4 (e.g., using SPR or ELISA) to confirm specificity and rule out other potential pathways. This is critical to validate the mechanistic hypothesis.

Thank you for your kind comment. The inhibitory effect of 225RANKL295 on osteoclast activation and the binding affinity with LGR4 were clarified at current published article (01 OCT 2025). So we added this reference as following; 27. 27.            Jang, Y.; Cho,Y.J.; Ko, Y.; Lee, H.; Kim, B.; Lee, C.M.; Lim, W. LGR4 binding small RANKL fragment inhibits RANKL-induced bone resorption. Eur Cell Mater 2025, 53, 15-27, doi: 10.22203/eCM.v053a02.

Minor Concern:

1. In 3D simulations (Fig. 1d, 3c), label key residues (e.g., 295 position) to enhance clarity and highlight the structural basis of the mutations.

Thank you for your kind comment. The 3D simulation in Figure 1D has been oriented to clearly show the key Pro295 region and highlighted as magenta spheres. Figure 3C also has been oriented to clearly show the 295 location and displayed as spheres.

Round 2

Reviewer 2 Report

Comments and Suggestions for Authors

No more comments.